# The Role of Online Learning Environments in the Enhancement of Language Learners' Intercultural Competence: A Scoping Review of Studies Published between 2015 and 2022

Barbara Muszyńska [1] , Joanna Pfingsthorn [2,*] and Tim Giesler [2]

[1] Faculty of Applied Studies, University of Lower Silesia, 53-611 Wrocław, Poland; barbara.muszynska@dsw.edu.pl
[2] Faculty 10, University of Bremen, 28359 Bremen, Germany; giesler@uni-bremen.de
[*] Correspondence: pfingsthorn@uni-bremen.de

**Abstract:** Developing intercultural competence (IC) through foreign language is believed to lead to rejecting prejudices and stereotypes and fostering bilingualism and biculturalism. Despite the growth of publications on technologies and IC, a significant gap exists between what is known (evidence) and what is done (practice) at the levels of decision making and course design. This scoping review, guided by the PRISMA Extension for Scoping Reviews guidelines, is conducted to systematically map peer-reviewed literature, taking a longitudinal perspective to update the existing reviews, identify knowledge gaps, and provide a new conclusion to the topic investigated. A transparent, replicable review protocol was designed a priori. A formal Advisory Group was established to incorporate various perspectives and ensure the applicability of the review findings. The main findings suggest that the concept of IC is not uniformly defined across the studies examined, and the development and dynamic nature of the concept is not captured. Numerous studies rely on chosen aspects of the construct only. Still, most of them report largely positive findings concerning the development of IC in FL online learning environments. It is possible that this high number of positive findings includes some cases of type II error or false positives.

**Keywords:** intercultural competence; ICC; foreign language education; online learning (environments); EFL; scoping review; empirical studies

## 1. Introduction

Technologies are seen as an integral part of today's learning process. They intensify multimodal possibilities in the act of meaning-making in different ways (Dahlström 2021) within and beyond classrooms (Leu et al. 2017; Psotka 2012) through exciting approaches such as reverse teaching, concept mapping, gamification, storytelling (Ergashev et al. 2021; Goodyear et al. 2019; Anwar et al. 2022). They also add a dimension of space and time by connecting students in virtual exchanges (synchronous or asynchronous) and can enhance the development of multimodal communication, intercultural competence (Lambert 1994), learner autonomy, and language development (Helm 2015). Developing intercultural competence through language is believed to lead to rejecting prejudices and stereotypes and fostering bilingualism and biculturalism (Fantini 2012).

Despite the growth of publications on technologies and intercultural communication, a significant gap exists between what is known (evidence) and what is done (practice) at the levels of decision making and course design. The literature on the use of online learning environments presents mainly teacher–student experiences (practice). Çiftçi and Savaş' (2018) narrative literature review of 17 studies (2010–2015) on the role of telecollaborating in language and intercultural learning discusses participants' views, technologies, and challenges. The study findings present the experiences of intercultural learning through telecollaboration (practice) and observable patterns but do not focus on its function or

purpose, or on the provision of implications for practice, which is why it is hard to gain a more comprehensive view of the researched topic. This may be due to the limitations in research methodology.

Scoping reviews are mostly used in health- and medical-related studies (Pollock et al. 2021). This may be due to the fact that in education, there are fewer studies that claim to be *systematic* (García-Gutiérrez et al. 2021). Nevertheless, the number of scoping reviews in social sciences and humanities is gradually growing, e.g., Han and Røkenes (2020) on a flipped classroom, García-Gutiérrez et al. (2021) on scoping reviews in environment and sustainability education, Hynes et al. (2022) a scoping review of literature on online international student collaboration in occupational therapy education.

The objective of a scoping review in this study is to systematically map relevant peer-reviewed research literature, taking a longitudinal perspective of studies published between 2015 and 2022 to update the existing review, identify knowledge gaps, and provide a new conclusion to the literature combining what is known with what is done and to comprehensively map the evolving landscape of the topic investigated (Peters et al. 2020; Arksey and O'Malley 2005) in order to generate ideas and (re)direct research and development activities (McMahan and McFarland 2021) into the future, also to inform policy and practice development.

There are many definitions and models of intercultural competence (IC) (e.g., Stern 1983; Koester and Olebe 1989; Gudykunst 1994; Byram 1997; Hampden-Turner and Trompenaars 2000). In this study, intercultural competence is understood in line with Lambert's (1994) definition, where IC comprises five components: (1) world knowledge, (2) foreign language proficiency, (3) cultural empathy, (4) approval of foreign people and cultures, and (5) the ability to practice one's profession in an international setting. Named by Lambert, components of IC explicitly specify *foreign language proficiency*, which is crucial for the analysis in this study. Moreover, other components, like *knowledge*, are consistent with other IC definitions. Other categories also correspond with other definitions; for example, those listed as *skills* refer to elements 2 and 5 of Lambert's definition, and those listed as *attitudes* to elements 3 and 4, which is also why this definition was chosen for this study.

This study is guided by the PRISMA Extension for Scoping Reviews (PRISMA-ScR) guidelines (Tricco et al. 2018). The PRISMA-ScR (PRISMA extension for Scoping Reviews) was developed according to published guidance by the EQUATOR (Enhancing the Quality and Transparency of Health Research) Network for the development of reporting guidelines.

A scoping review approach was chosen due to the broad nature of our research question.

The scoping review protocol was developed a priori to ensure transparency and to reduce duplication of work. The research team is international, based in Central Europe, and aware of potential culturally specific biases. The scoping review protocol and the involvement of a formal Advisory Group from outside of Central Europe are instrumental in identifying research bias in this study. The inclusion criteria of the sources of evidence used are provided with a rationale, and the literature search strategy is reported in a manner that allows replication. The main research question guided the development of specific inclusion strategies and the protocol and followed the Population (FL learners across sectors of formal education), Concept (intercultural competence), and Context (online learning environments) approach (Peters et al. 2020).

RQ: What is known from the literature about the role of interactive digital learning environments in the enhancement of (primary, secondary, and university) foreign language learners' intercultural competence?

The above research question aims to provide an overview and a map of the topic investigated, including author details, year of publication, country location, research methodology, and participants to identify the focus and purpose of the analyzed studies.

A formal Advisory Group of renowned international scholars in the field has been established not only for consultation and review (Stage 6 of the Scoping Review Frame-

work, Peters et al. 2020) since this step is often underdeveloped in the practice of scoping reviews (García-Gutiérrez et al. 2021), but also to incorporate various perspectives, enhance understandings of the literature and ensure effective communication of findings (García-Gutiérrez et al. 2021). The experts consulted the research protocol to ensure the rigor of the scoping review and to ensure that the data prioritized in the review are sufficient to answer the RQ.

The final part of this article summarizes the findings, objectives, study strengths and limitations, and potential implications and suggestions for further research and educational practice are presented.

## 2. Methodology

### 2.1. Scoping Review Protocol

This scoping review protocol pre-defines the objectives, methods, and reporting of the review and allows for transparency of the process. The protocol was refined as the reviewers progressed through the review. The protocol encompasses the criteria that will be used to include and exclude sources of evidence and to identify what data is relevant and ways of extracting and presenting the data (The JBI Manual for Evidence Synthesis, Aromataris and Munn 2020). The checklist used in the process of designing the protocol is in line with the PRISMA Extension for Scoping Reviews (PRISMA-ScR) (Tricco et al. 2018) (see Appendix A below).

### 2.2. Identifying the Research Question

The main research question is the following: what is known from the literature about the role of online learning environments in the enhancement of primary, secondary, and university foreign language learners' intercultural competence?

This study follows the Population (foreign/second language learners, high-school or university students), Concept (intercultural competence), and Context (online learning environments) approach (Peters et al. 2020).

### 2.3. Identifying Relevant Studies

This scoping review is guided by the PRISMA-ScR (PRISMA extension for Scoping Reviews) ((Tricco et al. 2018) framework for scoping reviews. To be eligible for inclusion in this scoping review, a publication had to be related to the role of online learning environments in the enhancement of primary, secondary, and university foreign language learners' intercultural competence. Peer-reviewed journal papers, including empirical studies, were included if they were published between the period of 2015–2022 as there are no reviews conducted post-2015 in the field of foreign language learning and IC; articles reviewed were written in English, involved foreign language learners, and identified a role of online learning environments in the enhancement of language learners' intercultural competence. The overarching concept of interest is intercultural competence and how it is developed in the context of institutionalized foreign language education. While the context of this scoping review can be described as 'open', as it is not limited to any form of digital interactive learning environment sources of evidence and applies to any contextual setting irrespective of their geographical location, it is restricted to formal education systems. The results of our study could be applicable to other digital institutional contexts.

Papers were excluded if they did not fit into the conceptual framework of this study, focused on preschool or adult learners in informal education, were written in languages other than English, or primarily focused on digital forms of learning that did not involve an interactive exchange between learners from different linguistic and cultural contexts. The following databases were used for searching the articles: Education Resources Information Center (Eric) and Scopus, due to their relevance in gathering published research in the field of intercultural FL education as shown in Table 1 below.

**Table 1.** Inclusion and exclusion criteria.

| Criterion | Included | Excluded |
|---|---|---|
| Databases | Eric, Scopus | Other databases |
| Time frame | 2015–2022 | Articles published outside of this time frame |
| Publication type | Online peer-reviewed articles; single studies | Books and book chapters, grey literature, conference proceedings, editorials, book reviews, study protocols |
| Methodology | Quantitative, qualitative, and mixed studies | N/A |
| Language | English | Other languages |
| Focus (concept and context) | Articles with a focus on the role of online learning environments in the enhancement of FL learners' intercultural competence | Articles focusing on other aspects of other competencies; interaction with machines only; no interaction; no explicit ICC or IC focus |
| Participants | Foreign language learners, primary, secondary, and university students in institutionalized educational settings | Articles focusing on younger or adult CLIL, FL learners or FL users outside institutionalized settings, or no FL |

The search strategy followed a three-step procedure, as indicated below.

### 2.4. A Three-Step Search Strategy

Step One: An initial limited search of the chosen online databases using the following keywords (intercultural competence, foreign language learning, online learning) for titles and abstract screening (performed by the three reviewers) to further develop the search strategy and search terms. At the end of the procedure, duplicates were to be removed and keywords reevaluated. The initial analysis involved more keywords; however, the reviewers conducted the revision and decided that the search words mentioned above brought the most results related to the role of online learning in the enhancement of foreign language learners' intercultural competence. Studies that used terms synonymous with online learning, such as 'telecollaboration' or 'virtual exchange', and discussed foreign language learners and their intercultural competence were included in this study.

Step Two: The reviewers searched across the chosen databases (the Education Resources Information Center (Eric) and Scopus) using the following search terms: intercultural competence, foreign language learning, and online learning. Title and abstract screenings of all eligible articles were to be conducted by all three reviewers independently, and a decision was made on whether the chosen studies should be included in the review or not.

Step Three involved all reviewers screening full texts and deciding which studies were suitable for inclusion and data charting.

### 2.5. Data Charting

The ultimate purpose of charting the data was to identify, characterize, and summarize research evidence on a topic, including the identification of research gaps (Nyanchoka et al. 2019).

A data-charting form was developed by the reviewers, as shown in Table 2, to determine which variables to extract. Data from eligible studies were charted using the data-charting form below. The data-charting form was managed using Google Forms online.

### 2.6. Synthesis of Results

The scoping review includes a narrative description of the article selection process accompanied by a flowchart of the review process (from the PRISMA-ScR statement) and a presentation of the evidence below. The research team decided on the best method for analyzing the charted variables of the collected data. The flow diagram presenting the steps undertaken is included in the Results section of the paper, together with the other findings.

**Table 2.** Data-charting form.

| Author and Year | Context and Form of Online Collaboration, (A)synchronous) etc. | Participant Profile | Methodology | Findings and Conclusions |
| --- | --- | --- | --- | --- |
| | | | | |
| | | | | |

## 3. Results

The search strategies were drafted and further refined through team discussions. The following search terms were used in Step One of the search strategy: intercultural competence, online learning, and foreign language learning. The search strategy was piloted to check the appropriateness of keywords and databases to ensure systematic and reproducible study selection and data charting and to avoid potential errors.

As shown in Figure 1 below, 320 studies were identified in the initial search process (Step 1), out of which 223 studies did not meet the inclusion criteria specified above and were excluded from the search. A total of 97 studies' titles and abstracts were screened in Step 2; duplicates were removed (n = 8). The authors took the possibility of conducting an additional search to double-check that they were not excluding any other valuable contributions of interest to this study. The team started the next phase of screening when 75% (or greater) agreement was achieved in Step Two (The JBI Manual for Evidence Synthesis, Aromataris and Munn 2020). The number of articles selected for Step Three was 89. Full-text screening allowed the researchers to remove 64 studies that did not meet the inclusion criteria specified above. The articles were checked a few times by all of the reviewers. In total, 25 studies remained suitable for inclusion and data charting. The files with the searches were stored online so that all of the reviewers had access to them at all times. Having an odd number of reviewers helped to resolve any discrepancies and discuss any arising issues to prevent bias (The JBI Manual for Evidence Synthesis, Aromataris and Munn 2020).

Figure 2 below shows the publication frequency of the articles per year.

The data-charting form was improved in Step Three and calibrated by the research team using the first five studies to ensure consistency and modified in iterations based on increased familiarity with the included studies, as shown in Table 3 below, and used to screen all selected articles.

**Table 3.** Data-charting form.

| Author and Year | Context and Form of Online Collaboration, (Asynchronous) etc. | Participant Profile | Methodology | Findings and Conclusions |
| --- | --- | --- | --- | --- |
| Öztürk and Ekşi (2022) | Virtual exchange project tool: Zoom | University students pre-service English teachers | Qualitative content analysis | The participants' interactions revealed that the critical cultural awareness dimension of ICC has been achieved throughout the virtual exchange project. |

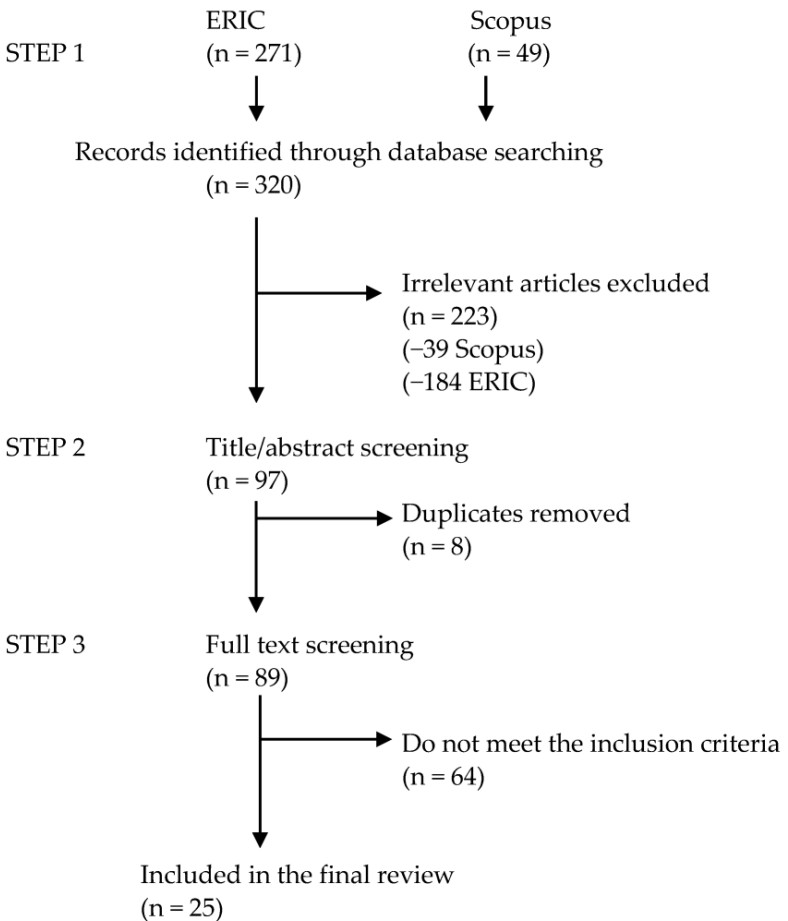

**Figure 1.** A three-step search strategy.

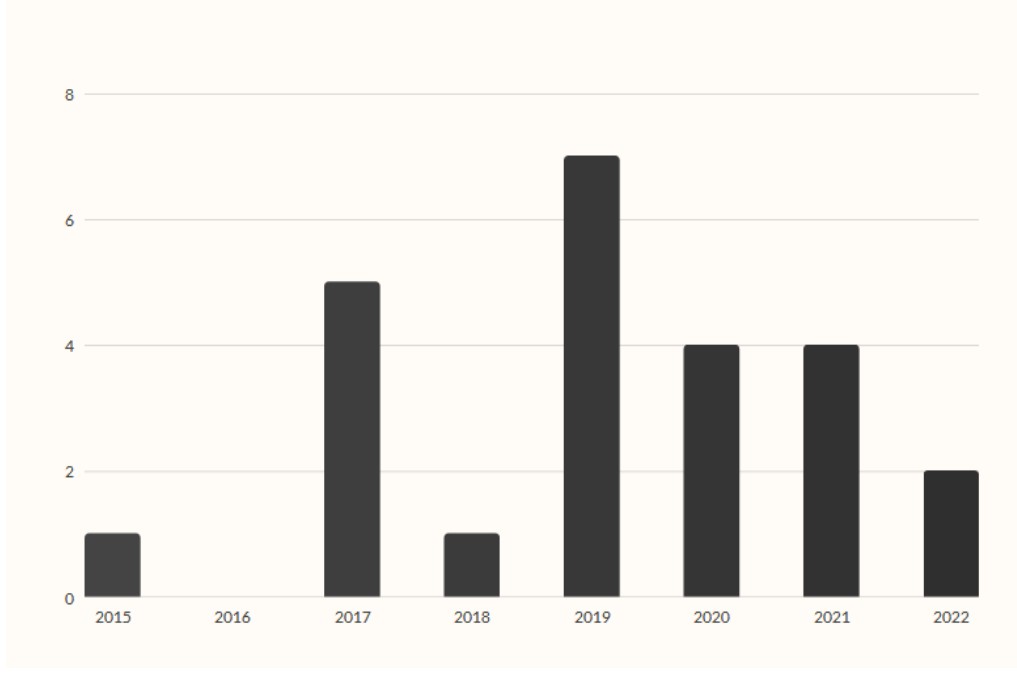

**Figure 2.** The frequency of the articles per year.

The authors have added another table during the synthesis of results that were not initially described in the Scoping Review Protocol but were helpful for the analysis, as presented in Appendix A.

## 4. Synthesis of Results

### 4.1. Online Learning Environments

Digital learning platforms are believed to enable self-directed learning, which is an important aspect of higher education (Sun et al. 2023). Nevertheless, only four articles reported on greater flexibility and customization of curricula based on the requirements of each student (57, 55, 6, 13), whereas the studies focusing on secondary and primary school (14, 16, 72) showed that teachers concentrated mainly on using online learning tools and technologies in an asynchronous mode to support students in developing an understanding of emerging technologies and self-motivation, which may prepare them for future education and work. Eleven studies showed that the online learning environments used allowed students to express themselves and collaborate on activities by sharing their thoughts and ideas (6, 7, 13, 38, 42, 47, 55, 57, 76, 88, 89).

In Avgousti's systemic review of ICC and online exchanges (Avgousti 2018), before the COVID-19 pandemic, the majority of the studies identified in the review were asynchronous, subsequently synchronous and asynchronous, and synchronous were listed last. Our review shows that synchronous and asynchronous exchanges are now used just as often. See Table 4 below.

**Table 4.** Modes of communication.

| Mode of Communication | Number of Studies |
|---|---|
| synchronous | 9 |
| asynchronous | 9 |
| synchronous and asynchronous | 5 |

According to the findings, the most commonly used tool for asynchronous communication was email (39, 57, 13, 14, 24, 6), followed by Moodle (72, 53, 16), Google Docs (80, 39, 57), Facebook, Messenger, social networking sites (82, 14, 47), and WhatsApp chat (6). Some technologies were used in both synchronous and asynchronous modes—TalkAbroad (76), WeChat, GG, and video chat (80, 42, 11, 14), Zoom (88, 4, 24, 6), Skype (89, 13, 47), and Edmodo (55). Five studies reported more uncommon technologies implemented—Voxpop (7), VR (34, 44), and Eliademy (38). Two studies did not report on the technology used (75, 36). In Avgousti's review (Avgousti 2018), email was also mentioned by authors as the number one tool used during their international projects.

The examined studies report on possible drawbacks of telecollaboration projects, such as the mismatch of partners' knowledge, foreign language skills (16), motivation (38), time management (89) also reported by Giralt and Murray (2019), time zones (38), students anxiety about grades which influences their performance (55), technical problems (89), insufficient instructions regarding the tasks to be performed (89) by native and non-native speakers in a project (89), and situating learners as passive observers and collectors of cultural information in IC contexts, rather than active contributors to their societies (57). The digital tools were used to provide international (a)synchronous exchanges as part of the face-to-face courses. Little space is devoted in the articles to the features and effectiveness of the online tools used in the projects.

The tasks in the studies under investigation were organized with the linguistic aim of improving FL skills and additionally IC(C) competence thanks to collaboration (4, 7, 11, 13, 16, 38, 42, 44, 53, 72, 75, 76, 80) as a result of an international project. The understanding of IC(C) in the papers varies from talking *about* the cultures (11, 24) to all ICC traits according to Byram (1997) (13, 36, 44, 55, 57, 76), but also in four categories (89). There were also studies that presented projects conducted in a foreign language with the ICC as the main focus (6, 44, 47, 55, 57, 82, 8, 89, 24).

### 4.2. Choice of Participants

Educational systems can be viewed as institutions since they mirror rules and conventions that provide social interactions among members of a population with a durable structure (Bowles 2006). As (sub)national regulations on how education is organized, standardized, and differentiated vary (Van de Werfhorst and Mijs 2010), it is essential to interpret the results of studies against their cultural and geographical backdrop as well as the size of the populations they focus on.

Most of the studies included in this scoping review were conducted in higher education contexts: 21 out of the 24 studies included university students, four of which engaged pre-service teachers pursuing their degrees (4, 7, 55, 88). Only three studies were conducted among pupils in elementary and secondary schools (14, 16, 72) as shown in Table 5. These results are in line with Wang and Vasquez's (2012) review, yet they do not display the gradual increase in the incorporation of elementary and secondary school contexts into research studies mentioned by Avgousti (2018).

**Table 5.** Academic status of participants.

| Academic Status of Participants | Number of Articles |
| --- | --- |
| University students | 18 |
| University teacher trainees | 4 |
| Pupils | 3 |

Two studies stand out in terms of the number of participants included in the research projects. Focusing on pupils from elementary and secondary schools, Lee and Park (2017) demonstrate results for a sample of 236 pupils. Taskiran's (2020) study was conducted among a total of 125 university students. While three other studies (39, 53, 88) did not report on the exact number of included participants, the samples of the remaining studies ranged from 2 (57) to 71 (31) and averaged 30 participants.

The location of the participants is presented in Table 6 below. The most frequent geographical location of participants involved in the studies was Asia, with 17 research projects anchored in countries such as Japan, Korea, Taiwan, China, Russia, Thailand, Indonesia, and Korea. Twelve studies involved participants from the USA. Ten studies included participants from various European countries such as Germany, Denmark, Spain, Finland, Italy, Hungary, and Bulgaria. Projects taking place in Australia and New Zealand were presented in three studies, as were three investigations from Latin America. Participants from the Middle East (Oman, Turkey, and Saudi Arabia) were involved in nine of the reported studies. This distribution of participants only partially resembles Avgousti's (2018) finding that the location where the most telecollaborative exchanges were conducted was America. While participants from North America were indeed involved in quite many projects, nine studies were conducted in settings that relied on English as a lingua franca of the participants.

**Table 6.** Location of participants.

| Location of Participants | Number of Studies |
| --- | --- |
| Europe | 10 |
| North America | 12 |
| Asia (incl. Russia) | 17 |
| Australia and New Zealand | 3 |
| Middle East (incl. Turkey) | 9 |
| Latin America | 3 |

### 4.3. Operationalization of Constructs and Methodological Aspects

The interpretation of results reported in empirical studies should also be contingent upon the methodological choices and the operationalization of constructs chosen by the authors. Studies included in this scoping report relied on quantitative and/or qualitative data collection procedures. While qualitative research "involves data collection procedures that result primarily in open-ended, non-numerical data which is then analyzed primarily by non-statistical methods", quantitative research "involves data collection procedures that result primarily in numerical data which is then analyzed primarily by statistical methods" (Dörnyei 2007, p. 24). Qualitative procedures were included in 19 studies, whereas 11 studies relied on quantitative data collection methods. Only seven studies adopted a mixed-methods approach, which, according to Dörnyei (2007, p. 24), "involves different combinations of qualitative and quantitative research either at the data collection or at the analysis levels". These results are only somewhat in line with Avgousti's (2018) review, which revealed a prevalence of qualitative studies and studies that adopted mixed methods. Avgousti (2018, p. XX) points out that the scarcity of quantitative studies could be attributed to the assumption that "the complexity of the mere skill of intercultural competence merits qualitative examination". While this explanation is plausible, the accessibility of validated quantitative instruments such as the questionnaire developed by Fantini and Tirmizi's (2006) ICC questionnaire, the Intercultural Effectiveness Scale, or the Intercultural Sensitivity Scale (ISS) developed by Chen and Starosta (2000), the Intercultural Communication Competence Questionnaire (ICCQ) by Matveev and Merz (2014) offers a practical solution for conducting quantitative investigations. Such tools were used in the studies by 14, 36, 47, 44, and 82.

Depending on its operationalization, the construct of ICC can be investigated through explicit and implicit measures. Explicit measures generally imply introspective awareness of the construct under investigation (Greenwald and Lai 2020), e.g., self-reports of participants or reflection. In contrast, implicit measures do not necessarily allow or require participants to be aware of the construct that is being assessed. In this sense, implicit measures are indirect measures of constructs, whereas explicit measures assess constructs in a direct way. An example of implicit measures of participants' ICC is linguistic analysis of discourse strategies or moves, length of text, or the choice of themes addressed in communication. Our scoping review revealed that explicit measures of ICC, in which participants relied on their introspection to evaluate the construct, were included in 13 studies. Implicit measures such as linguistic analysis of discourse choices and moves such as examples of expressions of curiosity about interlocutors' cultures, questions leading to (dis)confirming of existing beliefs, initiating, responding, and continuing moves, as well as their linguistic realizations (e.g., 75), were a part of 13 studies. On the one hand, providing opportunities for reflection on intercultural encounters is believed to foster critical thinking and confronting beliefs about cultural differences (e.g., Godwin-Jones 2003; Liddicoat and Scarino 2013). On the other hand, relying solely on participants' rumination runs the risk of illuminating only chosen parts of the underlying construct of interest. As Belz (2007) points out, assuming that ICC can be identified solely based on positive comments about intercultural encounters or with the apparent adoption of the interlocutor's values and/or viewpoints potentially is a short-sighted view. Ideally, investigations into ICC should try to approach the construct from multiple methodological perspectives and operationalize it not solely on introspection.

The scoping review revealed some variation in terms of the extent to which ICC was operationalized in reference to an established theoretical framework or a model. In relying on, e.g., Byram's (1997) model, Bennett's (2017) model, the functions of particular language forms, moves, and discourse strategies, or established rubrics such as the AAC&U (The American Association of Colleges and Universities) (2009) VALUE Rubric, a vast majority of studies assumed a valid approach to the investigation of the relevant construct (e.g., 13, 14, 16, 24, 36, 39, 38, 44, 47, 53). Some studies adopted self-designed instruments that did not seem to be validated or pretested (13, 29, 42).

Other studies focused on selective themes in their empirical part that do not reflect the chosen construct in its entirety. Oksana and Ruzana (2021, p. 549), for example, set out to investigate plurilingual competence, which they define as "communicative competence to which all knowledge and experience of language contributes and in which languages inter-relate and interact", yet they operationalize the concept primarily in terms of knowledge about the partner university and partner country; previous experience in online international learning; level of interest in the subject; level of interest in the project; perceived importance of the exchange; expectations; and the post-experience motivational value and reflection of learning. Similarly, Salih and Omar (2021) assume that intercultural competence can be operationalized as the ability to "observe any similarities between their culture and the other culture".

*4.4. Reported Challenges*

Not all studies reported challenges associated with the role of online learning environments in the development of ICC. Challenges that were reported included the following areas:

**Intercultural challenges:** Study (4) reports that it is challenging to "neutralize" students' attitudes regarding certain cultural constructs. The interaction between the two groups seemed to have amplified the perception of cultural differences. Similarly, (16) mentions that some participants display beliefs in cultural superiority to the extent that made the project coordinators willing to give up the endeavor. In (38), a similar thought is expressed, which says that "In this way, the project provided the participants with invaluable benefits and can be considered as a real blessing; on the other hand, it should not be regarded as an easy process." (p. 218).

In (55), the point of teachers' intervention and its potential positive impact on the development of ICC is raised. They underlie that "there were instances in which external intervention might have expanded the conversation or led it toward more fruitful directions, such as asking a follow-up question at a point that could open up a critical conversation" (p. 167). As (57) points out, "the intercultural learning outcomes of the interaction are principally dependent on the students themselves and their ability to explore and analyze cultural differences with their partner. This can often lead to a rather superficial analysis and understanding of difference" (p. 481).

**Engagement of participants:** Study (55) reports that participation in telecollaboration was a required component of the course, which had significance for participants' course grades. While, according to the authors, the participants involved in the project were encouraged not to worry about their grades, the status of an assignment may have influenced the interaction of the participants. The analysis included in other studies managed to capture only some of the participants; e.g., (75) reports that only American studies filled out a questionnaire at the end of the interaction and not their German counterparts. In a similar vein, (5) reports leaving out the opinions of students who did not complete the project, thus omitting potential problems of the project that acted as "deal-breakers" for those groups. In addition, (72) and (89) raise the issue of limited language proficiency, which can act as a communication barrier and lead to lower participation of individuals on the content level.

**Methodological issues:** Study (42) raises the question of the importance of control groups in settings in which new educational ideas are tested. Most studies do not include control groups in their designs, which makes the examination of developmental processes that potentially hinge upon online learning somewhat problematic. While it is true that the complex nature of such interaction implies a wide number of variables that are beyond the control of researchers, having no reference frame for the observed results makes their interpretation too absolute. Study (24) addresses the issue of the role of the cultural background of the participants, who, in their case, came from Asia. They suggest that participants following different rules of politeness and face-saving/face-threatening strategies could display different results.

**Practical issues:** Study (89) reports on a number of practical issues involved in online learning projects, including poor internet connection, difficulties finding a suitable time and date, punctuality, or lack of clarity in instructions.

## 5. Discussion

Web 2.0 technologies (wikis, podcasts, blogs, etc.) are believed to help learners generate content, collaborate with others, assess each other's work, and move toward peer learning. As a result, student learning can potentially become more active and engaged (Haleem et al. 2022). Online learning environments are said to recognize individual needs, incorporate technology into classroom instruction, and track student progress (Watty et al. 2016). Some of the studies examined in this review confirm what Mkrttchian et al. (2021) and Islam et al. (2021) found, namely, that teachers can use technology to create a more dynamic and exciting learning environment.

Studies focusing on secondary and primary school students concentrated mainly on using online learning tools and technology in an asynchronous mode. This supports students in developing an understanding of emerging technologies and self-motivation, which may prepare them for future education and work. Such a role of online learning environments was also described by Xazratov (2021), Hernandez-de-Menendez et al. (2020), and Li et al. (2021).

According to O'Dowd (2011), telecollaboration provides learners with cultural learning, a kind of learning that is not found in traditional textbooks, as information from their partners is subjective and not factual and objective. In the examined studies, students had the opportunity to treat the online learning environments as spaces for expressing themselves and collaborating on activities by sharing their thoughts and ideas, which is in line with Aleksandrov et al. (2012) and Abass et al. (2021).

The digital tools used in the examined studies indicate that telecollaboration is an area of 'product innovation' as described by Smith and Giesler (2023), i.e., "new or significantly different products and services" (Vincent-Lancrin et al. 2019, p. 21) that can potentially complement and extend a learning process both quantitatively and qualitatively. In quantitative terms, the digital tools used are a practical help in inviting intercultural communication partners synchronously or asynchronously into the classroom. Thus, intercultural exchanges can become more frequent and/or accessible, or perhaps in some cases—if learners cannot travel or can only travel with great difficulty—only made possible in this way. These rather practical effects seem to be based on a much greater availability and spread of corresponding digital tools (such as ZOOM or other video chat software) since the COVID-19 pandemic. The question of to what extent these practical affordances guarantee qualitative improvements, i.e., the extent to which virtual exchanges contribute to better intercultural understanding than other types of learning—remains unclear.

A more thorough investigation of the increased quality of online learning is further complicated by at least three factors that became clear during the evaluation: (1) The concept of intercultural competence is not uniformly defined across the examined studies, although there is an orientation towards globally known models such as Byram's, we find that a good number of the studies analyzed rely on chosen aspects of the construct only. Following Lambert's (1994) definition of IC, those would be either world or culture-specific knowledge, cultural empathy, or approval of foreign people and cultures. In addition, some of the examined studies focus on the participants' rumination, which (although being a source of helpful insights) offers partial and subjective access to the underlying construct. Furthermore, studies included in the review were not longitudinal in nature, i.e., they do not manage to capture the development and dynamic nature of the concept.

There are also potentially strong biases in (2) the choice of participants. Researchers often have easier access to their own (higher education) learners than to school pupils. This might be an explanation for why a good number of the studies are based on university students. In terms of many factors, such as motivation, cognitive development, or world knowledge, these groups of participants may not always be comparable. This implies that

the results of these empirical studies cannot be generalized to larger populations and need to be interpreted with an explicit reference to the involved sample type.

Given that most of the studies involved in this scoping review report on largely positive findings concerning the development of intercultural competence in online learning environments, it is necessary to consider whether there are truly few to no negative effects of online learning environments on the development of ICC, or whether this high number of positive findings includes some cases of type II error or false positives. Some research suggests that preexisting experiences and images of teaching that individuals acquire as pupils—or what Lortie (1975) refers to as an apprenticeship of observation—can be quite inflexible, persistent, and resistant to change. This phenomenon seems to be resistant to the influence of teacher education programs (e.g., Haritos 2004). It is thus plausible that some researchers firmly believe in the approach that they engage their participants in, which can partly overshadow firm scientific objectivity and lead to a confirmation bias. It would be desirable to share accounts of online learning experiments that failed or delivered "no" results.

Scoping reviews and the metadata that they provide can contribute to further developing empirical research in the field of language education and, as in this case, to taking a closer look at how 'innovative' online learning environments (like telecollaboration) affect a commonly used but not very well-defined concept (like intercultural competence); meta-studies can help practitioners and researchers to identify desiderata in a better and empirically validated way. Nevertheless, they certainly also help to bring the limitations and biases of one's own field into sharper focus and thus to identify methodological weaknesses and, if possible, to improve them in the long term by reflecting upon limitations of existing research.

Only four articles reported on greater flexibility and customization of curriculum to enable self-study and self-directed learning in an online learning environment, which contrasts with what Dudar et al. (2021) and Kosaretsky et al. (2022) found paramount in higher education.

**Author Contributions:** Authors contribution was equal. All authors have read and agreed to the published version of the manuscript.

**Funding:** This research received no external funding.

**Institutional Review Board Statement:** Not applicable.

**Informed Consent Statement:** Not applicable.

**Data Availability Statement:** No new data were collected.

**Conflicts of Interest:** The authors declare no conflict of interest.

## Appendix A

**Table A1.** Studies Reviewed.

| | Authors | Study Code | Participant No. | Countries Involved | EN as NL | EFL | Quantitative | Qualitative | Self-Report | Text Analysis | Mixed Methods | Solid Theoretical Framework | Operationalization Issues |
|---|---|---|---|---|---|---|---|---|---|---|---|---|---|
| 1 | Oksana and Ruzana (2021) | 4 | 20 | Spain, Finland | | 1 | 1 | | 1 | | | | 1 |
| 2 | Salih and Omar (2021) | 6 | 15 | Oman, USA | 1 | | | 1 | | 1 | | | 1 |
| 3 | Sardegna and Dugartsyrenova (2021) | 7 | 28 | Russia, USA | 1 | | | 1 | | 1 | | 1 | |
| 4 | Jung et al. (2019) | 11 | 51 | Korea, Japan, Taiwan | | 1 | 1 | 1 | 1 | 1 | 1 | | 1 |
| 5 | Hsu and Beasley (2019) | 13 | 71 | Taiwan, USA | 1 | | 1 | | 1 | | | 1 | |
| 6 | Lee and Park (2017) | 14 | 236 | USA, Australia | 1 | | 1 | | 1 | | | 1 | |
| 7 | Lázár (2015) | 16 | 78 | Turkey, Bulgaria, Italy, Hungary | | 1 | | 1 | | 1 | | 1 | |
| 8 | Freiermuth and Huang (2021) | 24 | 11 | Taiwan, Japan | | 1 | | 1 | | 1 | 1 | 1 | |
| 9 | Zheng et al. (2022) | 36 | 27 | USA, China | 1 | | 1 | 1 | 1 | | 1 | 1 | |
| 10 | Sevilla-Pavón (2019) | 39 | ? | Spain, USA | 1 | | 1 | 1 | 1 | | 1 | 1 | |
| 11 | Toscu (2021) | 38 | 24 | Turkey, USA | 1 | | | 1 | 1 | | | 1 | |
| 12 | Taskiran (2020) | 42 | 125 | Turkey, China | | 1 | 1 | 1 | 1 | | 1 | 1 | |
| 13 | Liaw (2019) | 44 | 20 | Taiwan | 1 | | 1 | 1 | 1 | 1 | 1 | 1 | |

**Table A1.** *Cont.*

| | Authors | Study Code | Participant No. | Countries Involved | EN as NL | EFL | Quantitative | Qualitative | Self-Report | Text Analysis | Mixed Methods | Solid Theoretical Framework | Operationalization Issues |
|---|---|---|---|---|---|---|---|---|---|---|---|---|---|
| 14 | Toscu and Erten (2020) | 47 | 31 | Turkey, USA | 1 | | 1 | 1 | 1 | | 1 | 1 | |
| 15 | Xu (2017) | 53 | ? | China, Japan, Thailand, Indonesia, Saudi Arabia, Australia | 1 | | | 1 | | | 1 | 1 | |
| 16 | Üzüm et al. (2020) | 55 | 48 | Turkey, USA | 1 | | | 1 | | | 1 | 1 | |
| 17 | O'Dowd (2020) | 57 | 2 | Spain, USA | 1 | | | 1 | | | 1 | 1 | |
| 18 | Tolosa et al. (2017) | 72 | 26 | Australia, New Zealand, Colombia | 1 | | | 1 | 1 | | | ? | |
| 19 | Ryshina-Pankova (2018) | 75 | 26 | Germany, USA | 1 | 1 | | 1 | | | 1 | 1 | |
| 20 | Tecedor and Vasseur (2020) | 76 | 18 | USA, Latin America | 1 | | | 1 | | | 1 | 1 | |
| 22 | Yang (2018) | 80 | 6 | USA, China | 1 | | 1 | | 1 | | | 1 | |
| 23 | Flowers et al. (2019) | 82 | 69 | Japan, Taiwan | | 1 | 1 | | 1 | | | 1 | |
| 24 | Öztürk and Ekşi (2022) | 88 | ? | Turkey, Germany, Israel | | 1 | | 1 | | | 1 | 1 | |
| 25 | Fernández and Pozzo (2017) | 89 | 44 | Argentina, Denmark | | 1 | | 1 | | | 1 | 1 | |

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
