# Peer review of "The Role of Online Learning Environments in the Enhancement of Language Learners’ Intercultural Competence: A Scoping Review of Studies Published between 2015 and 2022"

_languages, doi:10.3390/languages8030211_

Round 1
Reviewer 1 Report
The paper addresses a pertinent and current topic, contributes to the update of existing literature and identifies their shortcomings in order to provide relevant conclusions and improve future studies.
Scientific content is pertinent and methodology is accurate. The methodology is explained and the results are clear. References are appropriate and up to date.
Specific comments:
Please, make sure that in-text citations are homogeneous and adhere to the Journal’s guidelines. For example: “O’Malley, 2005” should have the comma removed, “Van de Werfhorst & Mijs 2010” should be Van de Werfhorst and Mijs 2010 or “(Dörnyei 2007: 24)” should be (Dörnyei 2007, p.24). Also review citation in line 448.
Likewise, the reference list should be homogeneous and adhere to the Journal’s guidelines.
There is a mistake in line 37: should be 2015.
Line 125: as ‘open’, as (comma after open to clarify idea).
Line 134: “The following databases” conveys the idea of several ones. However, there are two, so consider: The databases that were used for ….
Table 1: Review the alignment of the information in the "Included" column. It does not align with the first line and items of the other two columns.
Line 140: Perhaps this idea can be introduced or explained a little before getting on with the steps (much like what it was done with Data Charting, where there is a brief explanation after the subheading. (The search strategy followed a three-step.... or something on that line).
Line 221: Figure 2 has to follow the citation system in the heading.
Line 369 is missing the page for the direct quotation.
Line 391: the full stop should be after the bracket, like in line 399.
Line 405: 75 is missing the brackets?
Line 451: consider changing “this way” for another expression since the sentence ends with “in this way”.
Line 458: consider changing the hyphen for a comma.
Lines 485-6: there is a mistake in format.
Lines 493-495: consider removing some of the hyphens and use commas instead.
Line 497: Sentence should not start with But since there is a full stop before. Consider removing the full stop or using Nevertheless or However.
Author Response
Dear Reviewer,
Thank you for the review. We have now changed the in-text citations and added other changed indicated under the line numbers.
We've attached now the Word document with the changes made, the changes include also those changes requested by other reviewers.
With warm regards,
The Authors
Reviewer 2 Report
The article is both insightful and a valuable contribution to the field. The Authors have conducted a scrupulous study. There are only a few minor changes that I would suggest, which could enhance the precision of the text.
1. The authors state (see lines 71-72) that “The research team is international and thus potentially less blind towards culturally specific biases.”
Currently, there are numerous discussions within the field of intercultural communication and foreign language education about its Western-centric nature. Many argue that the field needs to undergo a significant shift in perspective and approach to become more inclusive and representative of diverse cultural contexts.
Instead of suggesting that the research team is "potentially less blind," I would recommend that the Authors explicitly acknowledge their geographic region of origin (for instance, Europe or Central Europe) and recognize the potential for cultural bias associated with it. This transparency will help mitigate any potential concerns about the impact of cultural perspectives on the research findings. Moreover, by openly acknowledging the geographic origin, the Authors will demonstrate a commitment to thorough and unbiased research, ultimately enhancing the article’s value and impact in the field.
2. Given that the Authors selected “online” as a keyword for their quantitative search, it would be advisable to maintain consistency by using "online" rather than "digital" in the research question (see line 78). This congruency in terminology will ensure clarity and alignment between the study's methodology and the framing of the research question.
3. The paragraph discussing the formal Advisory Group (lines 84-90) pertains to the Methodology and should therefore be relocated to Section 2 of the article. This adjustment will ensure that the content is appropriately placed within the relevant section, enhancing the article's structural coherence and making it easier for readers to understand the research process.
4. Could the authors shed light on how they navigated this potential challenge and if they took measures to account for such relevant publications?
In Step 1, the Authors conducted abstract screening using the specified keywords: "intercultural competence," "foreign language learning," and "online learning" (see lines 141-142). However, recent publications in this field often employ slightly varied terminology, often interchangeably, such as "virtual exchange," "telecollaboration," "telecollaborative project," "telecollaborative exchange," and so forth. I am curious about the Authors' process of re-evaluating the keywords and whether their chosen keywords inadvertently led to the exclusion of influential publications on the subject. Notably, numerous works authored by scholars like Melinda Dooly, Shannon Sauro, Ana Oskoz, and Melina Porto, among others, might have been unintentionally overlooked due to the keyword selection.
The description of "Step two" (lines 146-147) does not indicate any revision of the chosen keywords, which could potentially have led to relevant publications being missed. While it may not be feasible to address this shortcoming in the current study, it would be valuable for the Authors to consider this in future research endeavors and anticipate potential terminology variations to ensure a broader capture of relevant literature.
Furthermore, in Table 3, where "virtual exchange project" is cited as an example of context, it does raise questions about the potential impact of not including this specific phrase as a keyword. Given that this term is relatively frequently used in titles, it's possible that the screening results could have been different if it had been included in the list of keywords.
5. It is not clear why “learning” is missing from the list (see line 173)? If “learning” was selected as a part of the keywords, then the correct term should indeed be “foreign language learning” for consistency.
6. I wonder why Lambert’s (1994) definition of intercultural competence was chosen for this study? It's indeed important to provide reasoning for the selection of this particular reference and its relevance to the study's objectives.
7. In Table 7, I was puzzled by the fact that the Authors placed Russia in Asia. Russia is a transcontinental country that spans both Asia and Europe. The only Russian study referred to in the article is co-authored by a Russian researcher who is based in Moscow (the European part of Russian territory), and nothing in her study indicates that the study was conducted in Asia. The Authors may decide to put Russia in a separate category if they wish, but placing it in Asia would be a factual mistake that needs to be corrected.
8. There is a need for some minor corrections and for double-checking of references. For example, in lines 485-486 – there is a formatting issue with the word “apprenticeship” divided into two parts across two lines. Also, Lambert (1994) is missing from the List of references.
Overall, the article is very well-written and will definitely generate interest in the field. Congratulations to the Authors for putting this study together.
Author Response
Dear Reviewer,
Thank you for the review.
Re: 1) We have now addressed the potential Central European bias. Thank you for this comment.
Re: 2) Absolutely right, we have now changed to: online learning, but left digital tools.
Re: 3) We have decided to leave the short paragraph about the Advisory Group in the Introduction. We feel it fits best there. This decision is also based on other Scoping Reviews we've read.
Re: 4) We have made the reference to "virtual exchange," "telecollaboration," in Step 1 now. We have also addressed revision of the chosen keywords in Step 1, as this is when it took place.
Re: 5) We have added the word 'learning' now. Thank you.
Re: 6) We have now explained why why Lambert’s (1994) definition of intercultural competence was chosen for this study.
Re: 7) We are aware of the geographical location of Russia and Turkey, which is also placed in the Table in brackets, which is why these two countries are named in brackets. We have decided to do so, as not to create a separate category for the analysis.
Re: 8) We have thoroughly checked the References now.
We would like to thank you for this thorough review and comments that allowed us to improve the article.
With kind regards,
The Authors
We've attached now the Word document with the changes made, the changes include also those changes requested by other reviewers.
With warm regards,
The Authors
Reviewer 3 Report
Ambitious, well-structured and clearly substantiated, the article offers a thorough and academically sound scoping review of 25 peer-reviewed journal articles on the impact of online learning environments on the enhancement of students' intercultural competence. The arguments made by the team of researchers flow coherently and are backed up by a cornucopia of pertinent bibliographical references, woven into the architecture of the article with great dexterity. Likewise, the summary of the main findings and the conclusions drawn by the authors are most pertinent. In short, this review article represents a significant contribution to this specific field of studies.
Some minor typos/errors have been identified:
- Line 49: a word is missing in "Hynes et al. (2022) a scoping review".
- Lines 55-56: (Peters et al., 2020; Arksey & O'Malley, 2005). Commas should be deleted.
- Line 87: "but also to incorporate..." "Not only" should be added at an earlier point: "established not only for consultation and review .... but also to incorporate..."
- Line 91: "the findings, objectives study strengths and..." Please, revise this. Probably a comma is missing after "objectives".
- Lines 118-121: "... if they were: published... articles should be written in English; involve..." Please, revise the verbs used after "if they were".
- Lines 134-135 & Table 1. A minor inconsistency has been identified: Google Scholar is mentioned in the table but no in lines 134-135 as one of the databases used to conduct this research.
- Figure 1: "A Three-step search strategy". 'Three' should be written in lower case.
- Line 237: Whereas, as the studies focusing... teachers concentrated. Please, check punctuation. Maybe a comma should be inserted after (14, 16, 72).
- Line 264: "be performed (89) native". Is "by" missing before 'native'?
- Line 327: "review which revealed..." A comma should be added after 'review" (nondefining relative clause).
- Lines 384-422: third person singular -s missing in (4) report / (16) mention / (38) express / (55) raise / (55) report / 75 report / (5) report / (42) raise / (24) address / (89) report
- Line 402: "While according to the authors,..." A comma could be added after 'While'.
- Line 405: "e.g. 75 report" should read "e.g., (75) reports".
- Line 426: "(wikis, podcasts, blogs etc.)". A comma could be added before 'etc.'
- Line 426: consider replacing 'facilitate' with 'help'.
- Line 436: ", and self-motivation which may prepare..." Please, check punctuation.
- Line 444: "their thoughts and ideas which...". A comma should be added after 'which'.
- Line 457: "i.e. the extent..." A comma should be added after 'i.e.'.
- Line 465: "Following Lambert's (1994) definition of IC..." A comma should be added after 'IC'.
- Lines 485-486: please, revise line splitting.
- Line 488: (e.g. Haritos 2004). A comma should be added after 'e.g.'.
- Line 504: "higher education". A full stop is required at the very end.
- References: the titles of journals should be set in italics; references should be carefully revised to ensure they comply with the journal's guidelines.
Author Response
Dear Reviewer,
Thank you for the review. We have now changed the in-text citations and added other changed indicated under the line numbers.
We have also revised all of the References in the article.
We've attached now the Word document with the changes made, the changes include also those changes requested by other reviewers.
With warm regards,
The Authors